# Tuberculosis Predictive Index for Type 2 Diabetes Mellitus Patients Based on Biological, Social, Housing Environment, and Psychological Well-Being Factors

**DOI:** 10.3390/healthcare10050872

**Published:** 2022-05-09

**Authors:** Muhammad Atoillah Isfandiari, Chatarina Umbul Wahyuni, Agung Pranoto

**Affiliations:** 1Department of Epidemiology, Biostatistics, Population Studies and Health Promotion, Faculty of Public Health, Universitas Airlangga, Surabaya 60115, Indonesia; chatarina.uw@fkm.unair.ac.id; 2Department of Internal Medicine, Faculty of Medicine, Universitas Airlangga, Surabaya 60115, Indonesia; agung-p@fk.unair.ac.id

**Keywords:** diabetes mellitus, tuberculosis, predictive index, risk factors, biological factors, social factors, housing environment, psychological well-being

## Abstract

Indonesia is currently undergoing an epidemiological transition, with the double burden of disease due to increasing industrialization and urbanization leading to an increase in the prevalence of non-communicable diseases such as obesity and diabetes. On the other hand, the prevalence of infectious diseases such as tuberculosis remains high. Several factors were considered as risk factors in tuberculosis coincidence with type 2 diabetes mellitus. The purpose of this study was to develop a predictive index for tuberculosis in type 2 diabetes mellitus patients based on their biological, social, and environmental factors, and their psychological well-being as well. This case-control study involved 492 respondents consisting of 246 type 2 diabetes mellitus patients The variables studied were biological and social factors, the quality of their housing, and psychological well-being. Data analysis was conducted using a logistic regression test. The results showed that the predictive index formula was as follows: −3.218 + 0.867 × age + 1.339 × sex + 1.493 × history of contact with previous patient + 1.089 × glycemic control + 1.622 × tuberculosis clinical symptoms + 1.183 × body mass index + 0.891 × duration of diabetes mellitus + 0.454 × area of ventilation + 0.583 × psychological well-being. It is suggested that health workers, especially in primary health care facilities, will be able to increase the awareness of the risk of the coincidence of diabetes mellitus with tuberculosis.

## 1. Introduction

Tuberculosis is an infectious disease that infects about a quarter of the world’s population and is one of the leading causes of death worldwide [1]. Tuberculosis is caused by Mycobacterium tuberculosis and can be transmitted through the air. Currently, 86% of tuberculosis cases occur in the 30 highest TB burden countries, and Indonesia ranks third among countries with the highest prevalence of tuberculosis (8.4% of all tuberculosis cases in the world) after India (26%) and China (8.5%) [1]. García-Elorriaga G and Del Rey-Pineda G (2014) reported that one of the factors associated with the coincidence of tuberculosis in patients with type 2 diabetes mellitus is poor glycemic control (HbA1c > 7%) [2].

Moreover, low body mass index is a significant individual risk factor for new active tuberculosis incidence [3]. In a meta-analysis conducted by Patra et al. [3], BMI less than 18.5 kg/m^2^ in men contributed to 17% of tuberculosis cases in 14 countries with high tuberculosis burdens, and also contributed to 15% of tuberculosis cases in women in the same countries.

Clinical manifestations of tuberculosis appear when the immune status, including among diabetics, decreases [4]. People with weak immune systems as a result of chronic diseases such as diabetes have a higher risk of progressing from latent to active tuberculosis. One of the conditions that is frequently associated with the immune status of people with diabetes mellitus alterations is psychological well-being. Herawati et al., in their systematic reviews, showed that psychological well-being could increase the human body’s immune response, as shown by improvements in several indicators in saliva, blood, and plasma [5]. On the other hand, psychological ill-being was associated with decreased immune responses. 

Tuberculosis incidence can also be affected by the physical qualities of the house [6]. A house with a lack of sunlight and poor ventilation tends to create a moist and dark atmosphere, and this condition causes germs in the house, including Mycobacterium tuberculosis, which can last longer, i.e., from days to months [7]. The study in northern central Timor found that there is a significant relationship between house conditions, including the type of floor, ventilation, natural lighting, indoor temperature, humidity, and residential density of houses with incidence of tuberculosis [8].

At the same time, Indonesia is also experiencing an epidemiological transition, changing from infectious diseases problems to non-communicable diseases [9]. Communicable and non-communicable diseases contribute to the double burden of disease in Indonesia. One of the most prevalent non-communicable diseases in Indonesia is diabetes mellitus, with its prevalence continuing to increase rapidly, driven by changes in dietary patterns and lifestyle. Diabetes mellitus is a chronic, non-communicable disease that occurs when the pancreas does not produce enough insulin or cannot use the insulin effectively. The number of cases and the prevalence of diabetes has been increasing over a few decades [10]. In 2018, it was estimated that 0.4 million tuberculosis cases were related to diabetes globally [11]. Indonesia’s Ministry of Health reported that Indonesia ranked sixth in diabetes mellitus prevalence worldwide. 

In a meta-analysis study conducted by Wagnew et al., the prevalence of tuberculosis in diabetes was 4.72% (95% C.I. 3.62–5.83%), and in Asia it was 4.16% (95% C.I. 2.9–5.4%), making the double burden of disease consisting of tuberculosis and diabetes mellitus as a comorbidity an important but neglected emerging public health problem [12,13].

Several biological risk factors for tuberculosis can affect the comorbidity among diabetes mellitus patients, including age [14,15,16,17], sex [14,17], body mass index [17,18,19], and uncontrolled hyperglycemia [16,18]. Social factors, including economic status [18] and history of contact with previous tuberculosis patients are also well-known risk factors for tuberculosis [20], as are environmental factors, especially housing quality [8,18]. Moreover, the psychological well-being of the patients, which can also affect the immune status of diabetic patients, should be also considered a risk factor for tuberculosis comorbidity [21].

Therefore, developing a predictive index for tuberculosis incidence in diabetics based on these factors is important. The objective of this research is to develop a predictive index for tuberculosis in patients with type 2 diabetes mellitus, based on biological, social, environmental, and psychological well-being factors. The development of a predictive index for tuberculosis in type 2 diabetes mellitus patients based on these variables will be very beneficial because it will be useful for screening tuberculosis chance/risk of tuberculosis among type 2 diabetes mellitus patients so that the risk of tuberculosis developing among them can be reduced.

## 2. Materials and Methods

This research is a case-control study, conducted at 27 health centers and 1 pulmonary hospital in Surabaya, starting from May 2016 until April 2017, and involved 460 participants, consisting of 230 type 2 diabetes mellitus patients aged 18 to 70 years and undergoing the intensive phase of tuberculosis treatment as the case group, and 230 type 2 diabetes mellitus patients aged 18 to 70 years and without tuberculosis as the control group. All participants were involved by using simple random sampling. The ethical approval of this research was given by the Health Research Ethics Committee, Faculty of Public Health, Airlangga University, with statement of ethical approval number 547-KEPK.

Patients involved in the case group must be diagnosed with a new case of tuberculosis at least 3 months after type 2 diabetes mellitus diagnosis. Patients with immunocompromised status, including HIV infection or AIDS, post-transplantation status, chemotherapy, autoimmune diseases, those under immunosuppressant agent therapy, and tuberculosis relapse patients were excluded. 

Variables analyzed were biological indicators, including age, sex, glycemic control, clinical symptoms of tuberculosis complained of, subjective complaints related to complications of diabetes mellitus, body mass index, and duration of diabetes mellitus type 2. We also included social indicators, consisting of knowledge, attitudes, and actions related to tuberculosis prevention, family history of tuberculosis, history of intensive contact with previous tuberculosis patients, and history of high-risk tuberculosis transmission jobs. The other factors were the quality of the household environment, including the occupation density, ventilation width, indoor sunlight intensity, indoor and outdoor humidity, type of house floor, and indoor pollution, i.e., kitchen smoke pollution. Lastly, we analyzed the psychological well-being of the respondent as well.

Clinical complaints related to complications of diabetes mellitus, level of knowledge, attitude, and actions toward tuberculosis prevention data were obtained by interview using a questionnaire; glycemic control data were obtained through medical record study; and the psychological well-being of the respondent was measured using the WHO-5 psychological well-being index questionnaire. Validity and reliability tests in Indonesian people were carried out by Sarfika et al. (2021) [22]. This is among the most widely used questionnaires to assess subjective psychological well-being [23], which contains 5 positively phrased items as follows: (1) ‘I have felt cheerful and in good spirits’, (2) ‘I have felt calm and relaxed’, (3) ‘I have felt active and vigorous’, (4) ‘I woke up feeling fresh and rested’ and (5) ‘My daily life has been filled with things that interest me’ [23].

The quality of the housing environment was measured based on the Indonesian Healthy House Card criteria compiled by the Ministry of the Republic of Indonesia (1999), whereas the air humidity was measured using a hygrometer, light intensity was measured using a lux meter, and the indoor and outdoor temperature was measured using a thermometer.

Inferential statistical analysis used a multiple logistic regression test. We analyzed data in three steps to construct a predictive index formula for tuberculosis. The first step was selecting several variables to be included as candidate variables to be analyzed using multiple logistic regression. The criteria of the variables included as candidate variables were whether the variable had a significant correlation with tuberculosis and diabetes mellitus comorbidity, with a *p*-value < 0.25. 

The second step was to analyze the candidate variables with multiple logistic regression analysis, but we chose variables that had a significant correlation with tuberculosis and diabetes mellitus comorbidity, with a *p*-value < 0.05, to be variables for constructing a statistical model as a predictive index for tuberculosis in people with diabetes mellitus. 

The third step was to conduct an ROC (Receiver Operating Characteristic) measurement to determine the cutoff coordinate with the optimum sensitivity and specificity of the predictive ability of the formula.

## 3. Results

We initiated the construction of this predictive index by analyzing the relationship between 17 variables considered as risk factors for tuberculosis and diabetes mellitus comorbidity, with the occurrence of tuberculosis in the diabetic patients involved (Table 1). 

Based on simple logistic regression analysis, it was shown that among all 17 variables analyzed, there were only 14 variables with *p*-value < 0.25 available to be selected as candidate index variables. Furthermore, these variables were analyzed using multivariate logistic regression analysis into the index, i.e., the variables age, sex, contact history, family history of tuberculosis, glycemic control, the clinical symptoms of tuberculosis, subjective complaints related to complications of diabetes mellitus, body mass index, duration of diabetes mellitus, knowledge about tuberculosis, tuberculosis prevention-related action, smoking, quality places stay, and psychological well-being. In this case, the quality of the house environment variable is represented by the house ventilation width, considering that the other housing components listed on the Indonesian Healthy House Card by the Ministry of Health did not show a significant relationship with tuberculosis coincidence among diabetic patients.

Furthermore, we re-analyzed those 14 variables using multiple logistic regression tests to determine which variables fit the predictive index criteria by determining the *p*-value to be below 0.05. The test results are shown in Table 2. 

Considering that there were three variables with odds ratio values of less than one, those variables were removed from the analysis, and the results are that we had nine variables (Table 3).

According to the statistical analysis results shown in the table, now we can compile the predictive index formula for the possibility of tuberculosis occurrence among type 2 diabetes mellitus patients as follows:

Tuberculosis predictive index for type 2 diabetes mellitus patients =
−3.603 + 0.867 × age (≤55 y.o) + 1.339 × sex (male) + 1.493 × contact with previous tuberculosis patients (yes) + 1.089 × glycemic control (HbA1c > 7%) + 1.622 × clinical complaints resembling tuberculosis symptoms (>3 symptom) + 1.183 × body mass index (18.5 kg/m^2^) + 0.891 × duration of type 2 diabetes mellitus (10 year) + 0.454 × house ventilation width (10% house floor width) + 0.583 × psychological well-being (poor)

Furthermore, we calculated the probability of tuberculosis occurrence in type 2 diabetes mellitus patients based on the formula as follows:$$P=\frac{1}{1+e^{-Z}}$$

With Z = β0 + β1 x1 + β2 x2 + βn xn.

Annotation:
Pprobabilityβ0constant variable (Z value if x1 x2 ….. xn = 0)Zpredicted valueβnregression coefficienteconstant (2.714)x1 x2independent variables

Based on the tuberculosis predictive index for type 2 diabetes mellitus patients formulation above, if all of the variables are found in a type 2 diabetes mellitus patient, the Z value will be 5.918, so the probability of tuberculosis coincidence in that type 2 diabetes mellitus patient is be as follows: (1)$$P=\frac{1}{1+2.714^{-5.918}}$$
(2)$$P=\frac{1}{1+0.003}$$

P = 99.70%
(3)


According to the probability calculation result above, it can be concluded that if a type 2 diabetes mellitus patient is aged ≤ 55-years, male, has a history of contact with previous tuberculosis patients, has poor glycemic control (HbA1c > 7%), has more than 3 complaints resembling tuberculosis symptoms, has a BMI ± 18.5 kg/m^2^, suffers from type 2 diabetes mellitus ±10 years, has house ventilation width ± 10% of the house floor width, and experiences poor psychological well-being, then he/she has a 99.70% probability of experiencing tuberculosis coincidence. 

The next step was performing an ROC (Receiver Operating Characteristic) analysis. The result showed that upon an intersection coordinate of 0.405, this formula exhibited 82.9% sensitivity and 65.0% specificity, with 83.8% predictive capability according to the area under the curve (AUC) value (Figure 1).

These AUC values showed that the formula obtained had adequate power to predict tuberculosis coincidence in a patient with type 2 diabetes mellitus. Thus, with this sensitivity and specificity value, the intersection coordinate value 0.405 was determined as a predictive cutoff value. We can recalculate the sensitivity and specificity of this index using a 2 × 2 table if we compare the risk of tuberculosis and type 2 diabetes mellitus coincidence based on the index value compared to the actual diagnosis of the patients, as shown in Table 4 below:

From the 2 × 2 table above, we can calculate several values: the sensitivity, the specificity, the accuracy value of the diagnosis, PPV (positive predictive value), NPV (negative predictive value), and likelihood ratio.
(4)$$\mathrm{Sensitivity}=\frac{\mathrm{True}\mathrm{Positive}\left(\mathrm{TP}\right)}{\mathrm{True}\mathrm{Positive}\left(\mathrm{TP}\right)+\mathrm{False}\mathrm{Negative}\left(\mathrm{FN}\right)}$$
(5)$$\mathrm{Sensitivity}=\frac{204}{204+42}$$

Sensitivity = 82.9%
(6)

(7)$$\mathrm{Specificity}=\frac{\mathrm{True}\mathrm{Negative}\left(\mathrm{TN}\right)}{\mathrm{False}\mathrm{Positive}\left(\mathrm{FP}\right)+\mathrm{True}\mathrm{Negative}\left(\mathrm{FN}\right)}$$
(8)$$\mathrm{Specificity}=\frac{160}{86+160}$$

Specificity = 65.0%
(9)

(10)$$\mathrm{Accuracy}=\frac{\mathrm{True}\mathrm{Positive}\left(\mathrm{TP}\right)+\mathrm{True}\mathrm{Negative}\left(\mathrm{TN}\right)}{\mathrm{True}\mathrm{Positive}\left(\mathrm{TP}\right)+\mathrm{True}\mathrm{Negative}\left(\mathrm{TN}\right)+\mathrm{False}\mathrm{Positive}\left(\mathrm{FP}\right)+\mathrm{False}\mathrm{Negative}\left(\mathrm{FN}\right)}$$
(11)$$\mathrm{Accuracy}=\frac{204+160}{204+160+86+42}$$

Accuracy = 74.0%
(12)

(13)$$\mathrm{PPV}=\frac{\mathrm{True}\mathrm{Positive}\left(\mathrm{TP}\right)}{\mathrm{True}\mathrm{Positive}\left(\mathrm{TP}\right)+\mathrm{False}\mathrm{Positive}\left(\mathrm{FP}\right)}$$
(14)$$\mathrm{PPV}=\frac{204}{204+86}$$

PPV = 70.4%
(15)

(16)$$\mathrm{NPV}=\frac{\mathrm{True}\mathrm{Negative}\left(\mathrm{TN}\right)}{\mathrm{True}\mathrm{Negative}\left(\mathrm{TN}\right)+\mathrm{False}\mathrm{Negative}\left(\mathrm{FN}\right)}$$
(17)$$\mathrm{NPV}=\frac{160}{160+42}$$

NPV = 79.2%
(18)


We also calculated the PLR (positive-likelihood ratio) and the NLR (negative-likelihood ratio) values as follows: (19)$$\mathrm{PLR}=\frac{\mathrm{Sensitivity}}{1-\mathrm{Specificity}}$$
(20)$$\mathrm{PLR}=\frac{0.829}{1-0.65}$$

PLR = 2.377
(21)

(22)$$\mathrm{NLR}=\frac{1-\mathrm{Sensitivity}}{\mathrm{Specificity}}$$
(23)$$\mathrm{NLR}=\frac{1-0.829}{0.65}$$

NLR = 0.26
(24)


The PLR value shows that if the predictive index value is 2.37, the probability of a type 2 diabetes mellitus patient suffering from tuberculosis coincidence is higher. In contrast, the value of the NLR shows that if the predictive index value is 0.26, the probability of a type 2 diabetes mellitus suffering from tuberculosis coincidence is lower. 

Once the cutoff value was determined, we recalculated the Z value based on this cutoff value: (25)$$C=\frac{1}{1+e^{-z}}$$
(26)$$0.405=\frac{1}{1+e^{-z}}$$
(27)$$1+e^{-z}=\frac{1}{0.405}$$
(28)$$e^{-z}=1.469$$

−z = 0.385
(29)

z = −0.385
(30)


Interpretation:

<−0.385 = low tuberculosis–diabetes mellitus coincidence risk

≥−0.385 = high tuberculosis–diabetes mellitus coincidence risk

Thus, tuberculosis and type 2 diabetes mellitus coincidence prediction can be divided into two categories: If the index score is <−0.385, diabetic patients have low tuberculosis and type 2 diabetes mellitus coincidence risk;If the index score is ≥−0.385, diabetic patients have high tuberculosis and type 2 diabetes mellitus coincidence risk.

Now we simplify the formula, by determining that the cutoff index value as 0, so we added −0.385 to the model constant (−3.603). As the final result, with the cutoff value of 0, the new model constant now becomes −3.218. Thus, the final tuberculosis predictive index for type 2 diabetes mellitus patients formula, based on biological, social, and housing factors, and psychological well-being factors in this research are as follows:Tuberculosis predictive index on type 2 diabetes mellitus patients = 
−3.218 + 0.867 × age (≤55 y.o) + 1.339 × sex (male) + 1.493 × contact with previous tuberculosis patients (yes) + 1.089 × glycemic control (HbA1c > 7%) + 1.622 × clinical complaints resemble tuberculosis symptoms (>3 symptom) + 1.183 × body mass index (18.5 kg/m^2^) + 0.891×duration of type 2 diabetes mellitus (10 year) + 0.454 × house ventilation width (10% house floor width) + 0.583 × psychological well-being (poor)

## 4. Discussion

An index is the compilation of variables that are expected to be useful for evaluating state or status and allows the measurement of the citizenry against some of the changes that occur from time to time. A good index should meet these criteria: simple, timely, useful, measurable, and reliable. 

Based on those criteria, this index is expected to be useful for performing early screening to detect the probability of a type 2 diabetes mellitus patient developing tuberculosis so the prevention of tuberculosis in the patient can be implemented early also.

Some of the variables in this study were independently associated, theoretically, with the incidence of tuberculosis, so those variables were reasonably proper to build the tuberculosis predictive index for type 2 diabetes mellitus patients. 

The WHO-5 well-being index is a short questionnaire that consists of five simple statements to detect the subjective psychological well-being of the respondents [24]. This questionnaire had adequate validity as a screening tool and was implemented successfully in many studies [25,26,27,28,29]. Topp et al. (2015), in their meta-analysis, found that, from the 213 articles eligible to be reviewed, the WHO-5 well-being index had high clinometric validity as a screening tool for depression [23]. 

One of the limitations of this research is that glycemic control analysis was not measured based on HbA1c examination but by using the mean of three times fasting plasma glucose (FPG) levels and 2-h postprandial glucose (PPG) level examinations, respectively. Some of the variables with a strong evidence-based relationship with tuberculosis incidence, such as smoking habits, were not involved in the final index even though they exhibited a significant relationship to tuberculosis and type 2 diabetes mellitus coincidence in the previous analysis using simple logistic regression analysis. This matter could be caused by several factors, including cultural factors, since smoking habits are uncommon among females in Indonesian society and that the definition of smoking habits could be different between each person. According to that fact, this tuberculosis predictive index is optimally beneficial if it is implemented among non-smoking diabetic patients. 

To simplify the information, we can interpret the measurement result as follows: in terms of tuberculosis risk screening for type 2 diabetes mellitus patients, if the index score is <0, the patients have a low risk of developing tuberculosis coincidence, but if the index score is ≥0, the patient has a higher risk of developing tuberculosis coincidence.

With this predictive index, it is hoped that health workers, especially in primary health care facilities, will use it for examinations to further increase awareness of the risk of tuberculosis co-occurrence, especially for chronic patients who suffer from diabetes mellitus, and that it will also be used in an effort to prevent the early incidence of co-incidence. The results of the case finding based on this examination can be used by health workers to map the distribution of the coincidence of tuberculosis with type 2 diabetes mellitus in their area and can be used to search for the risk of transmission based on the environment.

Considering that body mass index is one of the risk indicators for the coincidence of diabetes mellitus with tuberculosis, health workers should not only treat diabetes mellitus but also assist patients in improving their immune status by increasing their body mass index, either through providing education about healthy foods with a balanced menu or through supplementary feeding programs that are suitable for people with diabetes mellitus and who are coincident with tuberculosis.

## 5. Conclusions

The tuberculosis predictive index formula to predict the risk of tuberculosis coincidence in type 2 diabetes mellitus patients was compiled as follows:−3.218 + 0.867 × age (≤55 y.o) + 1.339 × sex (male) + 1.493 × contact with previous tuberculosis patients (yes) + 1.089 × glycemic control (HbA1c > 7%) + 1.622 × clinical complaints resemble tuberculosis symptoms (>3 symptom) + 1.183 × body mass index (≤18.5 kg/m^2^) + 0.891 × duration of type 2 diabetes mellitus (≤10 year) + 0.454 × house ventilation width (≤10% house floor width) + 0.583 × psychological well-being (poor)
which can be interpreted as follows:if the index score is <0, the type 2 diabetes mellitus patient has a low risk of developing tuberculosis coincidence;if the index score is ≥0, the type 2 diabetes mellitus patient has a higher risk of developing tuberculosis coincidence.

It is expected that the health worker, especially in primary health care facilities, will be able to use this tuberculosis predictive index to screen the risk of tuberculosis coincidence in type 2 diabetes mellitus patients so that the risk of tuberculosis coincidence can be reduced.

## Figures and Tables

**Figure 1 healthcare-10-00872-f001:**
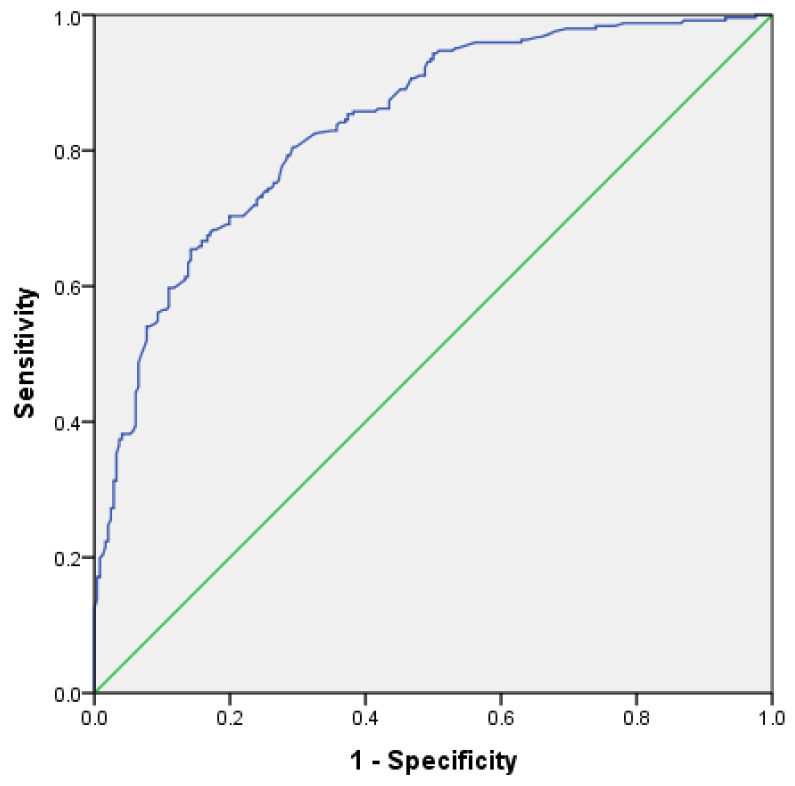
The area under the receiver operating characteristic (ROC) curve (AUC) for the prediction of tuberculosis in type 2 diabetes mellitus for the total sample.

**Table 1 healthcare-10-00872-t001:** Simple logistic regression test results before tuberculosis predictive index for type 2 diabetes mellitus compilation.

No	Variable	Sig.	Exp (B)	Result
1.	Age	0.001	2.414	Significant
2.	Sex	0.001	0.377	Significant
5.	Income	0.332	0.811	Not Significant
4.	Job-related to transmission risk	0.570	2.008	Not Significant
5.	Contact with previous tuberculosis patients	0.001	4.347	Significant
6.	Family history of tuberculosis	0.001	4.015	Significant
7.	Glycemic control	0.001	4.013	Significant
8.	Clinical complaints resembling tuberculosis symptoms	0.001	3.602	Significant
9.	Type 2 diabetes mellitus complications	0.001	0.450	Significant
10.	Body mass index	0.001	3.127	Significant
11.	Duration of type 2 diabetes mellitus	0.001	0.399	Significant
12.	Knowledge about tuberculosis prevention	0.003	0.578	Significant
13.	Attitude toward tuberculosis prevention	0.415	1.159	Not Significant
14.	Action toward tuberculosis prevention	0.027	0.665	Significant
15.	Smoking habits	0.001	2.942	Significant
16.	House ventilation width	0.010	0.623	Significant
17.	Psychological well-being	0.001	2.029	Significant

**Table 2 healthcare-10-00872-t002:** Multiple logistic regression test results before tuberculosis predictive index on type 2 diabetes mellitus compilation.

No.	Variable	B	Sig.	Exp (B)	95% C.I. for Exp (B)	
					Lower	Upper
1.	Age	0.847	0.000	2.332	1.475	3.688
2.	Sex	1.345	0.000	3.839	2.395	6.154
3.	Contact with previous tuberculosis patients	1.460	0.000	4.305	2.282	8.121
4.	Glycemic control	1.164	0.003	3.204	1.487	6.901
5.	Clinical complaints resembling tuberculosis symptoms	1.611	0.000	5.007	3.092	8.108
6.	Type 2 diabetes mellitus complications	−0.888	0.000	0.412	0.253	0.668
7.	Body mass index	1.208	0.007	3.346	1.399	8.006
8.	Duration of type 2 diabetes mellitus	0.872	0.004	2.392	1.323	4.324
9.	Knowledge about tuberculosis prevention	−0.544	0.020	0.580	0.366	0.919
10.	House ventilation width	−0.664	0.005	0.515	0.324	0.819
11.	Psychological well-being	0.694	0.015	2.001	1.146	3.495
	Constant	−2.936	0.000	0.053		

**Table 3 healthcare-10-00872-t003:** Multiple logistic regression test results before tuberculosis predictive index on type 2 diabetes mellitus compilation with statistically significant value.

No	Variable	B	Sig.	Exp (B)	95% C.I. for Exp (B)	
					Lower	Upper
1.	Age	0.867	0.000	2.380	1.510	3.750
2.	Sex	1.339	0.000	3.816	2.383	6.109
3.	Contact with previous tuberculosis patients	1.493	0.000	4.451	2.385	8.308
4.	Glycemic control	1.089	0.004	2.971	1.403	6.290
5.	Clinical complaints resembling tuberculosis symptoms	1.622	0.000	5.064	3.129	8.195
6.	Body mass index	1.183	0.006	3.263	1.398	7.616
7.	Duration of type 2 diabetes mellitus	0.891	0.003	2.437	1.355	4.380
8.	House ventilation width	0.454	0.046	1.574	1.008	2.457
9.	Psychological well-being	0.583	0.041	1.791	1.023	3.135
	Constant	−3.603	0.000	0.027		

**Table 4 healthcare-10-00872-t004:** Sensitivity and specificity of tuberculosis predictive index in type 2 diabetes mellitus.

Tuberculosis Predictive Index	Case (Tuberculosis–Diabetes Mellitus Coincidence Patient)	Control (Type 2 Diabetes Mellitus Patients)
	N (%)	N (%)
High coincidence risk	204 (82.93)	86 (34.96)
Low coincidence risk	42 (17.07)	160 (65.04)
TOTAL	246 (100)	246 (100)

## Data Availability

Not applicable.

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
