# Peer review of "Tuberculosis Predictive Index for Type 2 Diabetes Mellitus Patients Based on Biological, Social, Housing Environment, and Psychological Well-Being Factors"

_healthcare, 2022, doi:10.3390/healthcare10050872_

Round 1

Reviewer 1 Report

I am very grateful you for the invitation to review the manuscript healthcare-1689759 by Isfandiari and co-authors "Tuberculosis Predictive Index for Type 2 Diabetes Mellitus Patient Based on Biological, Social, Housing Environment, and Psychological Well-being Factors". The aim was to develop a predictive index of tuberculosis on Type 2 diabetes mellitus patients based on their biological, social, environmental factor, and their psychological well-being as well. The work is interesting but needs adjustments to increase the quality of the material.

Comments:

– Abstract: Include a brief problematic description of the study.

- Line 26-27: Change the repeated keywords by different words from the title to expand the search system.

- Page 1, Line 31-32: Check and update to Global Tuberculosis Report 2021.

- Introduction: Include definitions of tuberculosis and the total number of people affected by the disease, as well as the consequences (characterize the disease).

- Introduction:  In addition, authors should include in the Introduction a brief characterization of diabetes, as mentioned for tuberculosis.

- Introduction:  Authors should describe the importance of developing the index and how it can contribute to issues related to the two diseases;

- Page 4, Line 155: Check the sentence “candidate for index variables, consisting”.

- Inserting more references is recommended;

Author Response

Response to Reviewer 1 Comments

Thank you for giving us the opportunity to submit a revised draft of the manuscript “Tuberculosis Predictive Index for Type 2 Diabetes Mellitus Patient Based on Biological, Social, Housing Environment, and Psychological Well-being Factors” for publication in the Healthcare. We appreciate the time and effort you dedicate to providing feedback on our manuscript and thank you for your insightful comments and valuable improvements to our paper. We have included most of the suggestions made by reviewers. Please see below, in red, for point-by-point responses to reviewer comments and concerns. All page numbers refer to the revised manuscript file with tracked changes.

Point 1:  Abstract: Include a brief problematic description of the study.

Response 1: We thank the reviewer for pointing this out. We have added a brief problematic description in abstract (page 1, line 14-17)

Point 2:  Line 26-27: Change the repeated keywords by different words from the title to expand the search system.

Response 2: We’ve changed the keyword and added some words from the title of this manuscript (page 1, line 30-31)

Point 3:  Page 1, Line 31-32: Check and update to Global Tuberculosis Report 2021.

Response 3: We thank the reviewer for pointing this out. We have revised the data according to Global Tuberculosis Report 2021 (page 1, line 37-40)

Point 4:   Introduction: Include definitions of tuberculosis and the total number of people affected by the disease, as well as the consequences (characterize the disease).

Response 4: We’ve added a brief description about tuberculosis in the introduction (page 1, line 34-36)

Point 5:   Introduction:  In addition, authors should include in the Introduction a brief characterization of diabetes, as mentioned for tuberculosis.

Response 5: We’ve added a brief description about diabetes in the introduction (page 2, line 69-72)

Point 6:  Introduction:  Authors should describe the importance of developing the index and how it can contribute to issues related to the two diseases;

Response 6: We’ve added the importance of developing the index and how it can contribute to issues related to the two diseases (page 2, line 92-96)

Point 7:  Page 4, Line 155: Check the sentence “candidate for index variables, consisting”.

Response 7: We've changed “there were only 14 variables with a p-value<0.25 available to be selected as the candidate for index variables, consisting” to “only 14 variables with p-value<0.25 are available to be selected as candidate index variables” (page 4, line 182-184)

Point 8:  Inserting more references is recommended;

Response 8: We thank the reviewer for pointing this out. We’ve inserted more references in the revised manuscript.

We would like to thank the referee again for taking the time to review our manuscript.

Reviewer 2 Report

This is an interesting study aiming to generate an index to help physicians predict the risk of tuberculosis among people with type 2 diabetes.

My specific comments are as follows:

  1. Page 2, L 61-62: It would be more appropriate to say '... it was estimated that 0.4 million tuberculosis cases were related to diabetes globally' because the phrase 'were attributable to diabetes' implies that the only cause of TB was diabetes, which is certainly not the case.
  2. Page 3, L 114-117: Please provide an appropriate reference.
  3. Please define what do you mean by poor or good glycemic control. HbA1c <7%. According to all national and international guidelines, glycemic targets should be individualized according to patient characteristics (age, comorbidities, etc). For most patients however, the HbA1c target is < 7%.
  4. Page 6, L 216-222: The £ symbol needs to be replaced, it is not the correct one.
  5. In pages 9, L 314-317 and 10 L 322-326 the authors repeat themselves.
  6. Has the WHO-5 questionnaire been validated in the Indonesian population?
  7. Discussion: Please refer to potential preventive measures that a health care provider should take in the event of a high probability of TB in a patient with diabetes.

Author Response

Response to Reviewer 2 Comments

Thank you for giving us the opportunity to submit a revised draft of the manuscript “Tuberculosis Predictive Index for Type 2 Diabetes Mellitus Patient Based on Biological, Social, Housing Environment, and Psychological Well-being Factors” for publication in the Healthcare. We appreciate the time and effort you dedicate to providing feedback on our manuscript and thank you for your insightful comments and valuable improvements to our paper. We have included most of the suggestions made by reviewers. Please see below, in red, for point-by-point responses to reviewer comments and concerns. All page numbers refer to the revised manuscript file with tracked changes.

Point 1:  Page 2, L 61-62: It would be more appropriate to say '... it was estimated that 0.4 million tuberculosis cases were related to diabetes globally' because the phrase 'were attributable to diabetes' implies that the only cause of TB was diabetes, which is certainly not the case.

Response 1: We've changed “…were attributable to diabetes globally.” to “…were related to diabetes globally.” (page 2, line 73)

Point 2:  Page 3, L 114-117: Please provide an appropriate reference.

Response 2: We’ve added a reference regarding the selection of the questionnaire used (page 3, line 128)

Point 3:  Please define what do you mean by poor or good glycemic control. HbA1c <7%. According to all national and international guidelines, glycemic targets should be individualized according to patient characteristics (age, comorbidities, etc). For most patients however, the HbA1c target is < 7%.

Response 3: We thank the reviewer for pointing this out. We’ve clarified the poor glycemic control item by adding the description HbA1c >7%

Point 4:   Page 6, L 216-222: The £ symbol needs to be replaced, it is not the correct one.

Response 4: We've changed the “£” symbol to the correct one which is “±” (page 6, line 261-262)

Point 5:   In pages 9, L 314-317 and 10 L 322-326 the authors repeat themselves.

Response 5: We agree with reviewers that repeating the same sentence is ineffective. However, we thought that the conclusion section should contain the predictive index formula resulted from the study and how to interpret it. For this reason, we thought it will give better understanding if we leave it as it be

Point 6:  Has the WHO-5 questionnaire been validated in the Indonesian population?

Response 6: Validity and reliability tests for WHO-5 questionnaire were carried out by Sarfika et al (2021) (page 3, line 126-127)

Point 7:  Discussion: Please refer to potential preventive measures that a health care provider should take in the event of a high probability of TB in a patient with diabetes.

Response 7: We’ve added the potential preventive measures that the health care provider should take according to the result of this study (page 10, line 363-375)

We would like to thank the referee again for taking the time to review our manuscript.
